# Free Lunch: Frame-level Contrastive Learning with Text Perceiver for Robust Scene Text Recognition in Lightweight Models

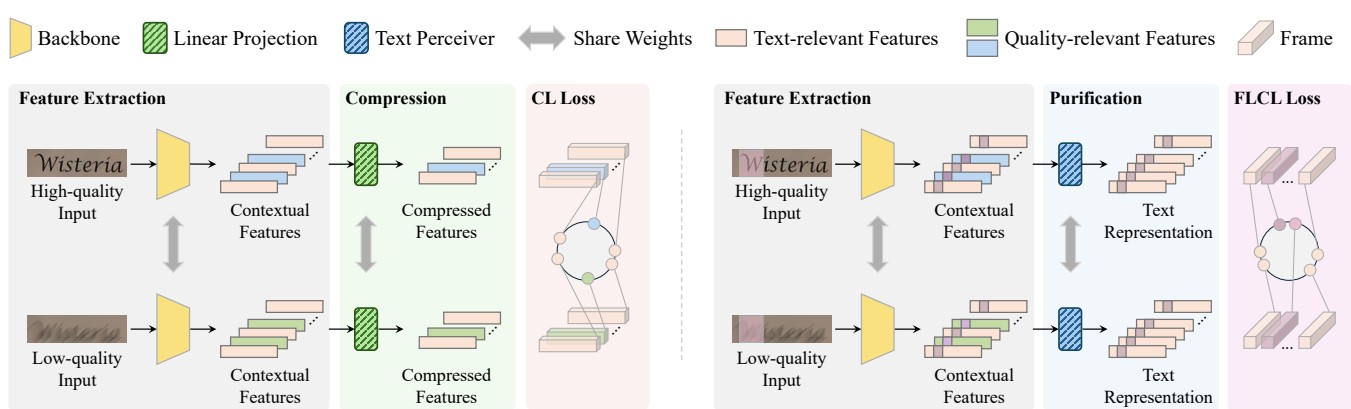

(a) Vanilla contrastive learning for STR

(b) Proposed frame-level contrastive learning for STR

**Figure 1: Comparison between vanilla contrastive learning and proposed frame-level contrastive learning.**

## ABSTRACT

Lightweight models play an important role in real-life applications, especially in the recent mobile device era. However, due to limited network scale and low-quality images, the performance of lightweight models on Scene Text Recognition (STR) tasks is still much to be improved. Recently, contrastive learning has shown its power in many areas, with promising performances without additional computational cost. Based on these observations, we propose a new efficient and effective frame-level contrastive learning (FLCL) framework for lightweight STR models. The FLCL framework consists of a backbone to extract basic features, a Text Perceiver Module (TPM) to focus on text-relevant representations, and a FLCL loss to update the network. The backbone can be any feature extraction architecture. The TPM is an innovative Mamba-based structure that is designed to suppress features irrelevant to the text content from the backbone. Unlike existing word-level contrastive learning, we look into the nature of the STR task and propose the frame-level contrastive learning loss, which can work well with the famous Connectionist Temporal Classification loss. We conduct experiments on six well-known STR benchmarks as well as a new low-quality dataset. Compared to vanilla contrastive learning

and other non-parameter methods, the FLCL framework significantly outperforms others on all datasets, especially the low-quality dataset. In addition, character feature visualization demonstrates that the proposed method can yield more discriminative character features for visually similar characters, which also substantiates the efficacy of the proposed methods. Codes and the low-quality dataset will be available soon.

## CCS CONCEPTS

• **Computing methodologies** → **Object recognition**; • **Computer systems organization** → Neural networks.

## KEYWORDS

Scene Text Recognition, Low-quality, contrastive learning, State Space Model

**ACM Reference Format:**
Anonymous Author(s). 2018. Free Lunch: Frame-level Contrastive Learning with Text Perceiver for Robust Scene Text Recognition in Lightweight Models. In *Proceedings of Make sure to enter the correct conference title from your rights confirmation emai (Conference acronym 'XX)*. ACM, New York, NY, USA, 10 pages. https://doi.org/XXXXXXX.XXXXXXX

## 1 INTRODUCTION

With the advancement of deep learning, robust Scene Text Recognition (STR) has emerged as a prominent topic in both academia and industry [55–57]. Numerous remarkable models have been proposed. It is evident that the scale of STR models is rapidly increasing. Additionally, iterative decoding is gradually gaining popularity thanks to its ability to achieve higher recognition accuracy, albeit at a significantly slower pace compared to methods based on Connectionist Temporal Classification (CTC).

However, text recognition serves as a fundamental module in practical document processing tasks, with limited resources allocated to this endeavor. Therefore, we need to utilize minimal resources to achieve maximal recognition performance. So we focus on the lightweight STR model in this paper.

Using little or no additional costs to improve performance has consistently been a popular approach. There are mainly two ways to achieve this. One involves employing more efficient loss functions such as FocalCTC [9], EnCTC [27], and DCTC [61]. The other entails adopting new training approaches, such as pluggable modules [37] during training, distillation learning, and Contrastive Learning (CL). Distillation learning necessitates a large and similarly-structured high-performance model as the teacher, which limits its applicability. In contrast, contrastive learning offers a more flexible and efficient usage. Some CL-based methods [28, 58, 59] have demonstrated success in STR tasks. However, most existing methods perform contrastive learning at word level, overlooking the fact that text recognition is actually a frame-wise task, which may limit effectiveness.

The complexity of existing features is also crucial for contrastive learning. Due to diverse image qualities, features extracted by CNNs or transformers often contain many irrelevant text features. This increases the difficulty of contrastive learning and diminishes final accuracy. Some methods utilize fully connected layers for feature projection in an attempt to mitigate the impact of irrelevant features. However, this approach is relatively direct and challenging for the purification of text-relevant information.

Based on these observations, in this paper, we propose a frame-level contrastive learning framework with a text perceiver for STR tasks, as illustrated in Fig. 1. The main difference compared to existing methods lies in conducting contrastive learning at the frame level. Traditional contrastive learning can only provide word-level statistical information, such as the number of frames containing the character 'w' or 'i' in the estimation result, but due to pooling operations, it cannot learn the exact frames. Our method addresses this issue by performing contrastive learning at each frame level without pooling, thereby achieving more accurate alignment while contrastive learning. Furthermore, in order to yield better text-relevant features, we design a bidirectional Mamba-based [13] Text Perceiver module to suppress text-irrelevant representations. We select nine well-known lightweight models and conduct experiments on six widely-recognized STR benchmarks as well as a specific low-quality dataset. All experiments demonstrate the effectiveness of the proposed method.

In summary, the main contributions of this paper are as follows:

(1) We propose a new frame-wise contrastive learning framework for scene text recognition task. It improves the performances of light-weight models without any new computational cost.

(2) We propose a new bi-direction Mamba-based module named Text Perceiver, which can purify the text-relevant information in the contextual features and make the outputs more closely related to the text content.

(3) We achieve new SOTAs on the lightweight STR models. Furthermore, we analyze the existing STR datasets and select the low-quality samples to form a new challenging dataset. This dataset is open access.

## 2 RELATED WORKS

### 2.1 Robust Scene Text Recognition

The robustness of STR models, specifically in low-quality scenarios, e.g., blur, low resolution, and noise, is a critical issue for applications. Many previous studies have explored the probability of enhancing the robustness of models in the wild, which can be divided into two categories. One of them aims to employ additional modules for preprocessing the low-quality inputs [5, 23, 37], where [23] proposes a text-specific hybrid dictionary for text image deblurring, [5] introduces a transformer-based text deblurring module, while [37] proposes a plugable super-resolution unit to improve the performance of the STR model faced with low-resolution text. On the other hand, with the development of language models, some work focuses on combining them with STR models to revise the incorrect prediction within low-quality contexts [8, 49, 50, 62]. These methods are effective, but they also introduce computationally heavy components, which are unaffordable for lightweight STR models. In this work, we propose a frame-level contrastive learning strategy for lightweight STR models to significantly enhance their performance in low-quality scenarios without any additional cost.

### 2.2 Contrastive Learning

Recently, [6, 12, 16] have significantly pushed the boundaries of representation learning by introducing contrastive learning. By generating positive samples via data augmentations and regarding other images as negative examples, [6, 16] pull together embeddings of positive pairs and push apart those of negative pairs. Additionally, [12] proves that merely using positive samples can also lead to a promising embedding for downstream tasks. [21] takes advantage of class labels as a criterion to separate positive and negative samples. For STR, [1] introduces a sub-word-level contrastive learning framework, in which patches from different visually augmented images are considered as positive samples. [29] proposes to view the same words in different semantic contexts as positive samples, thus deriving a word-level contrastive learning framework. [60] utilizes stroke-based partitions to help models focus on the topological structure of the stroke and learn text representations bottom-up. Existing contrastive learning-based STR methods employ linear projections to compress the contextual features, while it is still difficult for them to completely eliminate the influence caused by the text-irrelevant features. Different from them, we propose an efficient *Text Perceiver* instead of simple linear projections to achieve a more efficient purification of the text-relevant information in the contextual features. Additionally, we design a frame-level contrastive loss for STR models, which can improve their performance by providing more consistent supervision with the goal of the text recognition task.

### 2.3 State Space Model

For efficient long-range dependency modeling, [14] proposes a State Space Model (SSM)-based model, i.e., the Structured State-Space Sequence (S4) model, which is a novel alternative to CNNs or Transformers, and attracts further explorations due to its promising property of linearly scaling in sequence length. [45] proposes a new S5 layer by introducing MIMO SSM and efficient parallel

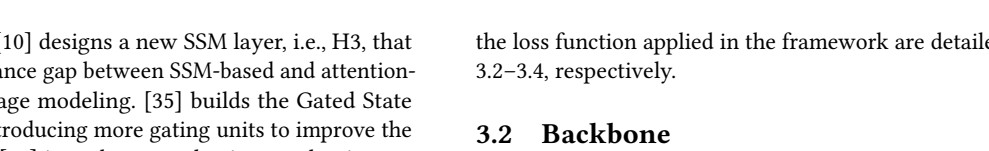

**Figure 2: The Architecture of proposed frame-level contrastive learning paradigm.**

scan into the S4 layer. [10] designs a new SSM layer, i.e., H3, that nearly fills the performance gap between SSM-based and attention-based models in language modeling. [35] builds the Gated State Space layer on S4 by introducing more gating units to improve the expressivity. Recently, [13] introduces a selection mechanism together with a specially designed hardware-aware algorithm into the SSM layer and builds a generic language model backbone, Mamba, which outperforms Transformers at various sizes on large-scale real data and enjoys linear scaling in sequence length. In this work, we explore the potential of the Mamba to purify the text-relevant features extracted by STR backbones and design a text perceiver to replace the linear projection employed in vanilla contrastive learning frameworks to improve their performance.

## 3 METHODOLOGY

### 3.1 Pipeline

The data pipeline of our proposed frame-level contrastive learning framework is shown in Fig. 2. Initially, high-quality inputs are subjected to generating the associated low-quality views via data augmentation, and then the high-quality inputs and their low-quality counterparts are separately fed into the backbone to extract the contextual features that are composed of task-required text features and quality-relevant image features. Subsequently, on the one hand, these contextual features are used to transcript the text via a decoder. On the other hand, we leverage a specifically designed text perceiver module to derive quality-invariant text representations from the contextual features, and then conduct the frame-level contrastive loss on the text representation space. The components and

the loss function applied in the framework are detailed in sections 3.2–3.4, respectively.

### 3.2 Backbone

The backbone of STR models generally consists of two components: a feature encoder, and an optional sequence model. There are three prevailing categories of feature encoders applied in the scene text recognition (STR) model. The first is CNN-based encoders, as exemplified by [8, 25, 43]. The second refers to transformer-based encoders, as demonstrated in [3, 7, 53]. The last integrates CNN with attention mechanisms, represented by [26, 51, 52]. Due to the difficulty of CNNs capturing long-range dependencies in sequences, the STR model with a CNN-based feature encoder often utilizes an extra sequence model to process the extracted visual features for better recognition accuracy. The most widely used sequence models include RNN [43], LSTM [11, 33], and transformer-based models [32, 39]. They convert visual features into contextual features that are used to transcript the text predictions via the decoder. As with vanilla contrastive learning, the proposed frame-level contrastive learning framework can be compatible with various backbones with different components, thereby facilitating flexible integration and showcasing substantial potential for applications. In the Experimental section, we have executed extensive experiments with diverse backbones to substantiate this adaptability.

### 3.3 Text Periceiver

*3.3.1 Motivation.* Contrastive learning is dedicated to allowing STR models to learn more discriminative text representations, thus

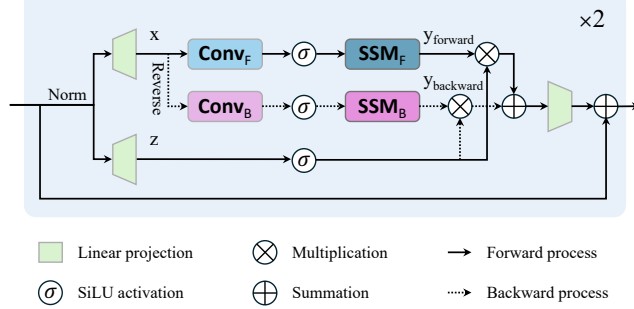

**Figure 3: The Architecture of the Text Perceiver.**

improving their recognition performance. However, when dealing with the text instance in a varying-quality context, the backbone will inevitably extract some quality-relevent features. In order to suppress the impact of these features on the training effect, vanilla contrastive learning frameworks commonly utilize several linear projections to compress the contextual features, and then calculate the contrastive loss in a more compact feature space. However, the low-quality samples may suffer from various distortions, which makes it difficult for simple linear projections to effectively perceive the text-specific information in the contextual features from different types of low-quality samples, resulting in suboptimal performance. To address this issue, we designed a SSM-based lightweight module, i.e., Text Perceiver, to replace the widely applied linear projection for more efficient purification of the text-specific information in the contextual features.

*3.3.2 Preliminaries.* The general SSM is inspired by the continuous system that maps a 1-D function or sequence $x(t) \in \mathbb{R} \mapsto y(t) \in \mathbb{R}$ through a hidden state $h(t) \in \mathbb{R}^N$:

$$\begin{aligned} h'(t) &= \mathbf{A}h(t) + \mathbf{B}x(t), \\ y(t) &= \mathbf{C}h(t), \end{aligned} \tag{1}$$

where $\mathbf{A} \in \mathbb{R}^{N \times N}$, $\mathbf{B} \in \mathbb{R}^{N \times 1}$, and $\mathbf{C} \in \mathbb{R}^{1 \times N}$ are separately the discretized evolution parameter and projection parameters. After the discretization via zero-order hold (ZOH) and parallelization, the SSM can be formulated as follows:

$$\begin{aligned} \overline{\mathbf{K}} &= (\mathbf{C}\overline{\mathbf{B}}, \mathbf{C}\overline{\mathbf{A}}\overline{\mathbf{B}}, ..., \mathbf{C}\overline{\mathbf{A}}^{M-1}\overline{\mathbf{B}}), \\ \mathbf{y} &= \mathbf{x} * \overline{\mathbf{K}}, \end{aligned} \tag{2}$$

where $\mathbf{x}$, $\mathbf{y}$ separately represents the input and output sequences. $\overline{\mathbf{A}}$ and $\overline{\mathbf{B}}$ are the discretized evolution parameter and projection parameter, respectively. As demonstrated by Eq. 2, the SSM exhibits the promising properties of linearly scaling in sequence length. However, it also illustrates the limitations of SSM in achieving input-dependent selection, which has been proven to be the key to the success of the attention mechanism.

To address this issue, Albert Gu proposes the Selective SSM, i.e., Mamba [13], which utilizes three linear projections combined with the discretization method to calculate the input-dependent $\overline{\mathbf{A}}_x$, $\overline{\mathbf{B}}_x$, and $\mathbf{C}_x$ and employs kernel fusion, parallel scan, and recomputation to improve the computational efficiency, allowing SSM to effectively yet efficiently focus on the important part of the inputs. Inspired

by the input-dependent selection mechanism and linear complexity of Mamab, we consider constructing a lightweight module based on selective SSM to replace the linear projection widely applied in vanilla contrastive learning for more effective purification of the text-relevant information in the contextual features.

*3.3.3 Architecture.* The original Mamba block is designed for 1-D sequence, which is inefficient for text recognition requiring spatial-aware understanding. Inspired by some applications of SSM in the vision task [42, 54, 63], we design the Text Perceiver, which adds an independent branch to process the reversed input for bidirectional feature extraction. The architecture of the proposed text perceiver is shown in Fig. 3. The input contextual feature is first normalized by the normalization layer. Subsequently, the normalized feature is separately projected to the feature $\mathbf{x}$ and the gated weight $\mathbf{z}$. For the feature $\mathbf{x}$, we process it from both the forward and backward directions. For each direction, we first employ a 1-D convolution to get the feature $\mathbf{x}'$. Inherited from Mamba, we utilize the $\mathbf{x}'$ to compute the $\overline{\mathbf{A}}_{\mathbf{x}'}$, $\overline{\mathbf{B}}_{\mathbf{x}'}$, and $\mathbf{C}_{\mathbf{x}'}$. Subsequently, we compute the $\mathbf{y}_{\text{forward}}$ and $\mathbf{y}_{\text{backward}}$ through the SSM layer. Finally, the $\mathbf{y}_{\text{forward}}$ and $\mathbf{y}_{\text{backward}}$ are gated by the weight $\mathbf{z}$ and added together to get the output.

## 3.4 Loss Function

There are two different-level loss functions in our framework, i.e., the recognition loss and the proposed frame-level contrastive loss. The former, similar to the previous works [34, 43], is used to provide a word-level supervision for STR models, while the latter is used to provide a character-level supervision for the STR models to learn quality-invariant text representations. Before delving into them, we first clarify the notations. For better performance, STR models are generally trained with large-scale synthetic datasets that are entirely composed of high-quality samples. Hence, given a batch of data $\{(\mathbf{X}_h^i, \mathbf{y}^i), 0 < i \leq N\}$ where $N$ is the batch size, their features are defined as $\{(\mathbf{Z}_h^i, \mathbf{U}_h^i, 0 < i \leq N\}$, where $\mathbf{Z}_h^i$ and $\mathbf{U}_h^i$ separately represents the text representation and the logit sequence. Similarly, after data augmentation, the associated low-quality views and their features are denoted as $\{(\mathbf{X}_l^i, \mathbf{y}^i, \mathbf{Z}_l^i, \mathbf{U}_l^i), 0 < i \leq N\}$.

*3.4.1 Recognition loss.* We compute recognition loss $\mathcal{L}_{\text{REC}}$ on logit sequences of both the high-quality and low-quality views, which can be formulated as:

$$\mathcal{L}_{\text{REC}} = \sum_{i=1}^{N} \mathcal{L}_{\text{CTC}}(\mathbf{U}_h^i, \mathbf{y}^i) + \mathcal{L}_{\text{CTC}}(\mathbf{U}_l^i, \mathbf{y}^i). \tag{3}$$

where $\mathcal{L}_{\text{CTC}}(\cdot)$ denotes the CTC loss [43] widely applied in the lightweight STR models.

*3.4.2 Frame-level contrastive loss.* Frame-level contrastive loss, i.e., $\mathcal{L}_{\text{FLCL}}$, aims to minimize the distance between each pair of associated frames in the projection sequence derived from the same text instance across different quality contexts, and maximize the distance between each pair of associated frames in the projection sequence derived from different text instances. Thus, giving a batch of paired projection sequences $\{(\mathbf{Z}_h^i, \mathbf{Z}_l^i), 0 < i \leq N\}$, the FLCL is formulated as:

$$\mathcal{L}_{\text{FLCL}} = \frac{-1}{NT} \sum_{i=1}^{N} \sum_{n=1}^{T} \log \frac{\exp(s(z_h^{i,n}, z_l^{i,n})/\tau)}{\sum_{m \in I_m} \exp(s(z_h^{i,n}, z_l^{i,m})/\tau)}, \tag{4}$$

where $z_h^{i,n}$, $z_l^{i,n} \in \mathbb{R}^{1 \times D}$ are the $n$-th frame of the projection sequence $\mathbf{Z}_c^i$ and $\mathbf{Z}_b^i$ respectively. $\tau \in \mathbb{R}^+$ is a temperature parameter, which is set to 1 in this work. $\mathbf{I}_m$ are the index set of all masked elements. $s(\cdot)$ is the cosine similarity which can be computed as $s(\boldsymbol{a}, \boldsymbol{b}) = \boldsymbol{a}^T \boldsymbol{b} / \|\boldsymbol{a}\| \|\boldsymbol{b}\|$. FLCL effectively facilitates aligning the representation of clear text instances and their low-quality counterparts at the frame level while also enhancing the extraction of discriminative features, which is pivotal for learning robust text representations.

Finall, the total loss takes the following form:

$$\mathcal{L}_{\text{total}} = \mathcal{L}_{\text{REC}} + \lambda \mathcal{L}_{\text{FLCL}}, \quad (5)$$

where $\lambda$ is a dynamically scaled scalar for balance between recognition loss and FLCL, which is computed as $\mathcal{L}_{\text{REC}} / \mathcal{L}_{\text{FLCL}}$ [30].

## 4 EXPERIMENTS

### 4.1 Datasets

All models are trained on a union of two commonly used synthetic datasets, i.e., **MJSynth** [17, 18] and **SynthText** [15], which contain about 14.4 M synthetic scene text images in total. Then, we first evaluate the models on six popular benchmarks: **IIIT5K-Words** (IIIT) [36] is the dataset crawled from Google image searches, which contains 3000 images for evaluation, and almost all of them are clear to recognize. **ICDAR2013** (IC13) [20] contains 857 images for evaluation, of which 9.3% have low quality. **CUTE80** (CT) is proposed in [41] for curved text recognition, where 288 testing images are cropped from full images by using annotated words, and about 9% of them are low-quality. **Street View Text** (SVT) [48] contains 647 outdoor street images collected from Google Street View, and about 14.2% of them have low-quality appearance. **Street View Text-Perspective** (SVTP) [40] is also cropped from Google Street View. There are 639 test images in this set, and about 20% of them are suffering from blurred or low-resolution distortion. **ICDAR2015** (IC15) [19] contains 1811 images for evaluation. The images are captured by Google Glasses while under the natural movements of the wearer, resulting in about 23.5% low-quality images. In addition, we provide a task-specific benchmark for evaluation: **Low-quality Text** (LQT) is made up of the low-quality samples collected from the previous six datasets, which have a total of 761 images. The details of the benchmarks are shown in Table 1.

### 4.2 Implementation Details

*4.2.1 Data augmentation.* We employ a combination of popular augmentation and visual distortions to generate low-quality views of the inputs, which can be formulated as follows:

$$\tilde{x} = f_2(f_1(x)), \quad (6)$$

where $x$ and $\tilde{x}$ separately denote the inputs and the associated low-quality samples. $f_1$ is the function for data augmentations, which includes *Curve, Stretch, Shrink, AutoContrast, Fog, Snow, Frost, Rain, Shadow.* $f_2$ is the function for visual distortions, which includes *GaussianBlur, DefocusBlur, MotionBlur, GaussianNoise, JpegCompression, Pixelate.* All of the operations are equal in probability and achieved by the *Straug* [2] library. Noteworthy, we have not conducted other augmentations on the high-quality inputs separately during the training phase.

**Table 1: Number and proportion of low-quality samples for evaluation benchmarks.**

| Benchmark | # of Low-quality smaples | # of total smaples | Ratio |
|---|---|---|---|
| IIIT | 6 | 3000 | 0.2% |
| IC13 | 80 | 857 | 9.3% |
| CT | 27 | 288 | 9.4% |
| SVT | 92 | 647 | 14.2% |
| SVTP | 131 | 639 | 20.3% |
| IC15 | 425 | 1811 | 23.5% |
| LQT | 761 | 761 | 100% |

*4.2.2 Base Model Selection.* To assess the generalizability of our proposed method, we have chosen nine popular light-weight OCR models for evaluation, which include CRNN [43], SVTR-T/S [7], EfficientNetV2-b0/b1 [46], EdgeViT-XXS/XS [38], and EfficientFormerV2-S0/S1 [24]. Notably, the EfficientNet series incorporates two Bi-LSTM layers with a hidden size of 256 for sequence modeling. Across all these models, a fully-connected layer is utilized as the decoder to transcribe contextual features into the text. In addition, the downsample ratio is set to [×32, ×4], the dimension of the output feature is set to 512, and the dimension of the contrastive learning feature is set to 192.

*4.2.3 Hyperparameters.* The rectification module [31, 44] is employed for distortion correction. All the input RGB images are resized to 32 × 100, and the maximum length of prediction is set to 25. We adopt the Adam optimizer [22] with a cycle learning rate from 2e-3 to 1e-8 for training, where the weight decay is set to 1e-5. The training batch size is 256, and the training epoch is 5. Gradient clipping is used at magnitude 5. All experiments are conducted on NVIDIA RTX 4090 GPUs.

*4.2.4 Evaluation Protocols.* We use word accuracy (ACC) to evaluate all models' performance, which is the ratio of the number of totally correct predictions over the number of test samples. Besides, we also report the number of parameters and the inference speed. Notably, only numbers and letters (case-insensitive) are evaluated.

### 4.3 Ablation study

To demonstrate the effectiveness of each component in the proposed framework, we perform an ablation study in this section. Since the IIIT, IC13, and CT include a small proportion of low-quality samples, we marked them as high-quality datasets, while the SVT, SVTR, IC15, and LQT are marked as low-quality datasets. For efficiency, we adopt the EfficientFormerV2-S0 trained by vanilla contrastive learning as the baseline on all seven datasets.

*4.3.1 Ablation on Key Components.* The proposed framework has two key components, i.e., text perceiver (TP) and frame-level contrastive loss (FLCL). The TP is designed to more efficiently purify the text-relevant information in the contextual features, while the FLCL is proposed to provide a character-level link between the text instances with different qualities. We conduct ablation to validate the effectiveness of TP and FLCL, and the results are shown in

**Table 2: Ablation on key components. 'LN' denotes linear projection, 'TP' denotes text perceiver, 'CL' denotes contrastive loss, and 'FLCL' denotes frame-level contrastive loss.**

| LN | TP | CL | FLCL | High-quality datasets | Low-quality datasets |
|----|----|----|------|------------------------|-----------------------|
| ✓ |    | ✓  |      | 87.2 | 72.4 |
|    | ✓  | ✓  |      | 88.5 | 73.7 |
| ✓ |    |    | ✓    | 88.3 | 72.9 |
|    | ✓  |    | ✓    | **90.0** | **74.6** |

**Table 3: Ablation on the architecture of Text Perceiver.**

| Bidirectional strategy | Recognition ACC. | |
|------------------------|------------------|---|
|                        | High-quality datasets | Low-quality datasets |
| None | 89.2 | 73.7 |
| Bidirectional Sequence | 89.4 | 74.1 |
| Bidirectional SSM | 89.5 | 74.4 |
| Bidirectional SSM + Conv | **90.0** | **74.6** |

Table 2. We can observe that applying TP to replace LN can bring an improvement of 1.3% on average accuracy. On the other hand, compared with popular word-level contrastive loss, FLCL results in an average accuracy improvement of 0.8%. Finally, it is worth noting that the combination of TP and FLCL further boosts the average accuracy of about 1.2%.

*4.3.2 Ablation on Text Perceiver.* Compared with Mamba [13], the proposed text perceiver adopts a special bidirectional strategy for more efficient feature extraction. To illustrate the effectiveness of this design, we perform an ablation on the design of text perceiver, where we consider these strategies:

- **None**. We directly adopt the Mamba block instead of the linear projection to purify the text-relevant information in the contextual features within the forward direction.
- **Bidirectional Sequence**. We randomly flip the contextual features during the training phase, which is like data augmentation.
- **Bidirectional SSM**. We add an extra SSM layer for each block to process the reversed contextual features.
- **Bidirectional SSM + Conv**. Based on Bidirectional SSM, we further add a Convolution layer before the SSM in the backward branch. (as shown in Fig. 3).

As indicated in Table 3, adopting the Mamba block achieves better performance than linear projection, while applying additional bidirectional strategies can further boost the averaged accuracy to varying degrees. (0.3%~0.8%). Noteworthy, the strategy of a bidirectional SSM layer with convolution achieves the best results, which demonstrates the effectiveness of the text perceiver.

*4.3.3 Ablation on scale of loss.* The relative scale of recognition loss and the frame-level contrastive loss will be changed at different epochs of the training process. Based on this observation, we

**Table 4: Ablation on the scaled scalar of the loss function.**

| Scaled scalar | Recognition ACC. | |
|---------------|------------------|---|
|               | High-quality datasets | Low-quality datasets |
| 1 | 89.5 | 74.4 |
| 0.5 | 89.7 | 74.1 |
| 0.2 | 89.9 | 73.9 |
| 0.1 | 89.4 | 73.4 |
| Dynamic | **90.0** | **74.6** |

consider a dynamic scaled scalar to balance different losses during the training. To verify the effectiveness of the dynamic scalar, we compare it with several static scales, and the results are assessed in Table 4. For static scalars, we can see that paying too much attention to contrastive loss will affect the recognition performance of the model faced with high-quality samples, while paying little attention to contrastive loss will make the performance of the model decline under low-quality scenarios. However, as for the dynamic scalar, it is able to provide the model with the highest recognition accuracy in both high-quality and low-quality datasets.

## 4.4 Results

*4.4.1 Model-wise comparison.* To demonstrate the effectiveness of the proposed framework, we compare the performance of it and the CTC framework with / without contrastive learning (CL) on seven popular light-weight OCR models mentioned in Sec. 4.2, and the results are reported in Table 4. We can clearly see that, compared with standard contrastive learning, our method can provide an average accuracy improvement of about 2% for various models with different backbones over all benchmarks without any additional cost, which profoundly verifies the effectiveness of our method at the model level. Overall, since it is difficult for vanilla contrastive learning to efficiently extract text-relevant information from the extracted contextual features, the models trained by CL are usually suffering from the unbalanced performance between the samples of different quality. However, due to the text perceiver, our framework can provide more consistent performance improvements for the models when faced with different-quality samples. To be specific, CRNN, the most classical, representative, and widely used light-weight text recognition model, obtains a 4.2% average accuracy increment via our framework. Besides, the advanced CTC-based text recognition method, the SVTR series, gains over 1% improvement in average accuracy with our method. In addition, all the rest of the text recognition models, i.e., EfficientNet, EdgeViT, and EfficientFormer series, also achieved about 1%~2.5% improvement in average accuracy by our method. Furthermore, for datasets containing a large number of blurred or low-resolution samples, e.g., SVT, SVTP, IC15, and LQT, the improvement brought by our framework is more significant.

*4.4.2 Comparisons with State-of-the-Arts.* To illustrate the superiority of our methods, we compare it to the state-of-the-art methods designed for the light-weight OCR models, e.g., FocalCTC [9], EnCTC [27], and DCTC [61], with the classic OCR models. They are

**Table 5: Results of Model-wise Comparison on seven benchmark datasets made up of different percentage of low-quality samples. Bold ACCs are the model-wise better results. 'CTC + CL' refers to adapting vanilla contrastive learning framework with CTC loss as the recognition loss to train the model.**

| Backbones | Methods | IIIT | IC13 | CT | SVT | SVTP | IC15 | LQT | Avg. | Param (M) | Time (ms) |
|---|---|---|---|---|---|---|---|---|---|---|---|
| CRNN | CTC† | 84.3 | 90.3 | 61.3 | 78.9 | 64.8 | 65.9 | 40.6 | 72.4 | | |
| | CTC + CL | 85.7 | 91.0 | 68.4 | 82.6 | 67.4 | 68.2 | 45.0 | $75.6_{(+3.2)}$ | 7.16 | 4.8 |
| | Ours | **88.2** | **92.0** | **76.7** | **84.5** | **72.6** | **73.5** | **50.3** | $\mathbf{79.8}_{(+7.4)}$ | | |
| SVTR-T | CTC‡ | 94.5 | 96.3 | 88.2 | 91.6 | 85.4 | 84.1 | 60.3 | 86.5 | | |
| | CTC + CL | 94.0 | 96.5 | 88.4 | 92.4 | 87.6 | 85.5 | 62.8 | $87.5_{(+1.0)}$ | 5.98 | 4.5 |
| | Ours | **95.0** | **96.8** | **90.5** | **93.6** | **88.3** | **87.1** | **64.8** | $\mathbf{88.9}_{(+2.4)}$ | | |
| SVTR-S | CTC‡ | 95.0 | 95.7 | 92.0 | 93.0 | 87.9 | 84.7 | 67.2 | 88.2 | | |
| | CTC + CL | 94.8 | 96.2 | 91.4 | 93.8 | 89.3 | 86.2 | 69.5 | $89.0_{(+0.8)}$ | 10.13 | 8.0 |
| | Ours | **95.2** | **96.8** | **92.8** | **94.5** | **90.2** | **87.8** | **71.4** | $\mathbf{90.1}_{(+1.9)}$ | | |
| EfficientNetV2-b0 | CTC | 88.9 | 94.9 | 69.8 | 85.5 | 74.0 | 73.9 | 52.3 | 82.1 | | |
| | CTC + CL | 89.0 | 95.0 | 72.2 | 85.0 | 74.5 | 75.0 | 54.4 | $82.9_{(+0.8)}$ | 6.95 | 4.9 |
| | Ours | **89.2** | **95.4** | **74.2** | **86.2** | **76.4** | **75.7** | **55.3** | $\mathbf{83.9}_{(+1.8)}$ | | |
| EfficientNetV2-b1 | CTC | 89.1 | 94.2 | 73.3 | 86.7 | 75.5 | 76.1 | 57.0 | 84.3 | | |
| | CTC + CL | 90.4 | 95.1 | 74.8 | 86.5 | 76.0 | 77.3 | 58.4 | $85.2_{(+0.9)}$ | 9.57 | 8.3 |
| | Ours | **91.2** | **95.7** | **75.5** | **88.2** | **77.8** | **78.4** | **59.3** | $\mathbf{86.3}_{(+2.0)}$ | | |
| EdgeViT-XXS | CTC | 88.7 | 93.8 | 72.6 | 86.5 | 75.5 | 76.1 | 52.1 | 83.4 | | |
| | CTC + CL | 89.0 | 93.5 | 73.0 | 87.6 | 77.2 | 78.8 | 54.3 | $84.5_{(+1.1)}$ | 5.95 | 4.5 |
| | Ours | **89.2** | **94.5** | **75.4** | **88.3** | **79.0** | **80.2** | **57.5** | $\mathbf{86.0}_{(+2.6)}$ | | |
| EdgeViT-XS | CTC | 90.3 | 94.5 | 76.7 | 86.6 | 78.2 | 77.1 | 54.3 | 84.6 | | |
| | CTC + CL | 90.5 | 94.0 | 77.5 | 87.8 | 79.3 | 78.4 | 57.0 | $85.6_{(+1.0)}$ | 8.62 | 8.2 |
| | Ours | **90.8** | **95.2** | **78.6** | **89.4** | **80.7** | **79.5** | **58.4** | $\mathbf{86.7}_{(+2.1)}$ | | |
| EfficientFormerV2-S0 | CTC† | 85.9 | 91.2 | 73.3 | 80.7 | 70.9 | 70.3 | 45.3 | 77.9 | | |
| | CTC + CL | 86.2 | 92.0 | 74.0 | 83.5 | 72.4 | 72.8 | 48.5 | $79.8_{(+1.9)}$ | 3.56 | 3.8 |
| | Ours | **87.5** | **93.4** | **75.4** | **85.9** | **76.0** | **76.6** | **53.8** | $\mathbf{82.3}_{(+4.4)}$ | | |
| EfficientFormerV2-S1 | CTC | 88.2 | 93.4 | 77.1 | 83.3 | 75.0 | 74.1 | 52.1 | 81.5 | | |
| | CTC + CL | 87.4 | 93.5 | 77.0 | 84.5 | 77.0 | 75.8 | 56.4 | $82.7_{(+1.2)}$ | 6.15 | 4.0 |
| | Ours | **88.6** | **93.9** | **79.5** | **87.0** | **77.4** | **77.5** | **58.2** | $\mathbf{84.2}_{(+2.7)}$ | | |

The results of † are reported by [4], and the results of ‡ are reported by [7].

widely applied in real-life scenarios to enhance the performance of the light-weight OCR model without additional cost. Since these methods are not specifically designed for low-quality text images, we only report the results on the six popular benchmarks for fair comparison, which are shown in Table 5. We can observe that in datasets containing a larger proportion of low-quality samples, i.e., SVT, SVTP, and IC15, our method can provide the models with significantly the best performance among SOTAs, illustrating its advantage in enhancing recognition performance in low-quality scenarios. Furthermore, although our method aims to enhance the recognition performance of the models in low-quality scenarios, it can also effectively enhance the model performance when faced with high-quality samples. In general, our method brings the largest increment of accuracy for not only CRNN but SVTR-T on most benchmark datasets, resulting in a 6.9%, and 1.9% improvement of the average accuracy, respectively, which is more than double the best of SOTAs.

## 4.5 Visualization Analysis

In a low-quality scenario, it is very difficult for models to identify samples with confusing characters. Our method introduces a concise yet effective text perceiver to replace the linear projection and suggests an additional frame-level distillation between high-quality samples and associated low-quality views besides the recognition supervision, which promotes the model to extract more discriminative features when faced with low-quality text images and thus improve its overall performance. To qualitatively demonstrate the effectiveness of our method, we provide a series of visualization analyses with EfficientFormerV2-S0 that is trained by vanilla contrastive learning, i.e., the baseline, and the proposed frame-level contrastive learning, respectively.

To verify the effectiveness of our method, we conducted a feature visualization study with t-SNE [47]. Specifically, we select several hard example groups that are composed of characters prone to being wrongly recognized as each other. We crop some examples

**Table 6: Comparison with the state-of-the-art methods, where the results of the DCTC are reported by [61]. Bold ACCs are the best results; Underline ACCs are the second best results.**

| Models | Variants | Venue | IIIT | IC13 | CT | SVT | SVTP | IC15 | Avg. |
|--------|----------|-------|------|------|-----|-----|------|------|------|
| CRNN | CTC | TPAMI'15 | 84.3 | 90.3 | 61.3 | 78.9 | 64.8 | 65.9 | 77.3 |
| | FocalCTC | Complexity'19 | 81.2 | 89.6 | 60.2 | 80.1 | 63.0 | 65.2 | $75.6_{(-1.7)}$ |
| | EnCTC | NeurIPS'18 | 85.6 | 90.1 | 59.0 | 81.5 | 62.9 | 64.7 | $77.1_{(-0.2)}$ |
| | DCTC | AAAI'24 | **88.9** | 90.7 | 68.1 | 82.4 | 65.4 | 66.1 | $\underline{79.9}_{(+2.6)}$ |
| | Ours | - | 88.2 | **92.0** | **76.7** | **84.5** | **72.6** | **73.5** | $\mathbf{84.2}_{(+6.9)}$ |
| SVTR-T | CTC | TPAMI'15 | 94.5 | 96.3 | 88.2 | 91.6 | 85.4 | 84.1 | 90.8 |
| | FocalCTC | Complexity'19 | 94.3 | 96.0 | 87.9 | 91.0 | 85.1 | 84.1 | $90.6_{(-0.2)}$ |
| | EnCTC | NeurIPS'18 | 94.5 | 94.9 | 88.2 | 90.8 | 85.4 | 84.3 | $90.6_{(-0.2)}$ |
| | DCTC | AAAI'24 | **95.4** | 96.4 | 89.9 | 92.3 | 86.1 | 85.3 | $\underline{91.7}_{(+0.9)}$ |
| | Ours | - | 95.0 | **96.8** | **90.5** | **93.6** | **88.3** | **87.1** | $\mathbf{92.7}_{(+1.9)}$ |

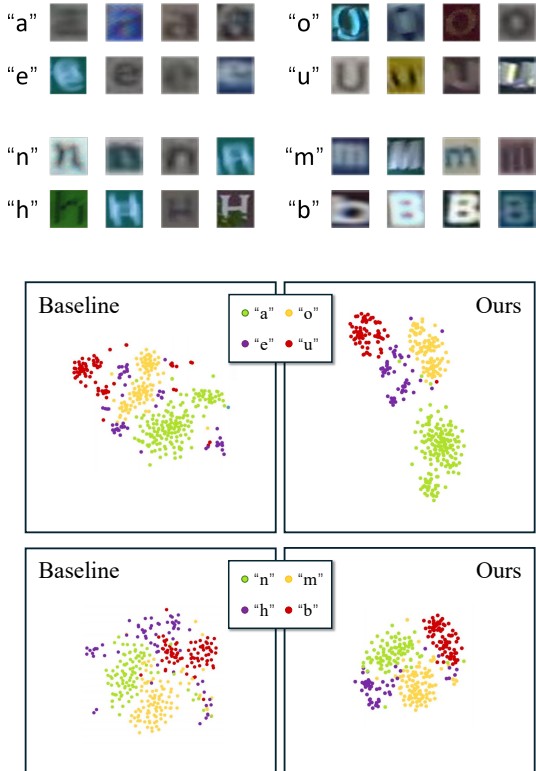

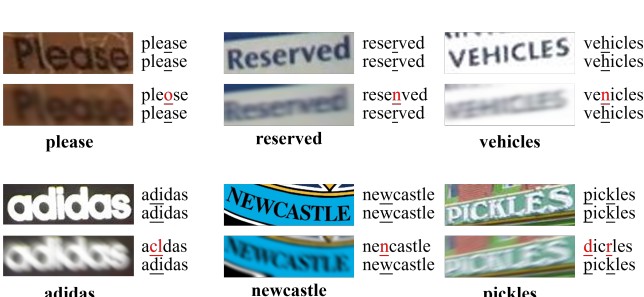

**Figure 5: Qualitative examples where the baseline fails but our method succeeds. From top to bottom are the predictions of the baseline and our method.**

appearances, our method can still drive the model to extract more discriminative features that are more cohesive than those extracted by the baselines. Furthermore, some predictions of high/low-quality sample pairs are given in Fig. 5. We can find that it is easier for the model trained by our method to distinguish the confusing characters in low-quality cases and make consistent predictions between high-quality and low-quality samples. For example, the prediction of the first low-quality example in Fig. 5 is corrected from 'pleose' to 'please' by our method.

**Figure 4: Feature visualization of the hard example groups. From top to bottom are separately the examples and the associated feature projections, where each row represents a group.**

from the images on the test sets and separately fetch their feature embeddings from the baseline and our method. Fig. 4 shows the feature projections of two hard example groups, where different characters are marked with different colors. We can clearly observe that even when faced with low-quality samples with very similar

## 5 CONCLUSION

In this paper, we propose a concise yet quite effective strategy to enhance the performance of lightweight STR models when faced with low-quality samples without additional cost, which includes a SSM-based text perceiver and a frame-level contrastive loss. By employing the text perceiver to derive the text-specific information from the contextual features extracted by the backbone and then prompting character-focused feature learning via frame-level contrastive loss, our method can help STR models learn more robust text representation, thus improving their recognition performance. The superiority of our method has been illustrated by both quantitative and qualitative analysis of several popular STR benchmarks. The proposed method not only has excellent generalization performance but also achieves the best results compared with SOTAs.

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
