# OpenReview forum: "Free Lunch: Frame-level Contrastive Learning with Text Perceiver for Robust Scene Text Recognition in Lightweight Models"
_acmmm.org/ACMMM/2024/Conference — MM2024 Poster_

### Official Review · Reviewer_8sNe · 2024-05-14

**Rating:** 3
**Confidence:** 4

**Summary:**

Lightweight models play an important role in real-life applications, especially in the recent mobile device era. However, due to limited network scale and low-quality images, the performance of lightweight models on Scene Text Recognition (STR) tasks is still much to be improved. The paper proposes a new efficient and effective frame-level contrastive learning (FLCL) framework for lightweight STR models. The FLCL framework consists of a backbone to extract basic features, and a Text Perceiver Module (TPM) based on a Mamba structure to focus on text-relevant representations. Experiments on six well-known STR benchmarks are conducted. Compared to vanilla contrastive learning, the proposed method can perform better on challenging low-quality datasets.

**Strengths:**

1) The paper proposes to perform frame-wise contrastive learning for scene text recognition.
2) The paper proposes to replace the linear project head with a mamba-based project head.
3) Experiments are conducted on different backbones for evaluation.

**Limitations:**

1) Frame-level contrastive learning is not new in scene text recognition which has been explored by SeqCLR [1]. However, neither reference nor comparison is represented for SeqCLR which is one of the most related works. Another contribution is the use of a mamba-based projection head, which is quite incremental. Besides, the parameter and inference speed comparison is also absent to demonstrate the lightweight nature of TPM.
[1] Sequence-to-Sequence Contrastive Learning for Text Recognition.
2) Lack of comparison of the most recent scene text recognition papers in MM or CVPR, which is not reasonable.
3) Detailed issues. In L327, the reference of [25] (DB, a scene text detection) is not appropriate; In Formula (4), the scope of the negative samples is not clear;
4) Given the description in Sec. 4.2.1 and Figure 2, after the data augmentation, the frame of the low-quality and the high-quality image features may be not spatially aligned. Doesn't it influence the process of frame-level contrastive learning?

**Suitability:**

3

---

### Official Review · Reviewer_vaCi · 2024-05-20

**Rating:** 3
**Confidence:** 4

**Summary:**

The paper proposes a novel framework to enhance the performance of Scene Text Recognition (STR) models, especially under poor image quality conditions. It introduces a frame-level contrastive learning (FLCL) approach combined with a Text Perceiver Module (TPM) to filter text-relevant features and suppress noise. This method improves the model's ability to recognize text by generating more discriminative features. The authors demonstrate the effectiveness of their approach through experiments on several STR benchmarks and a new low-quality dataset, showing substantial performance gains.

**Strengths:**

(1)	Traditional contrastive learning techniques in STR operate at the word level, which can overlook the sequential nature of text data. By implementing contrastive learning at the frame level, the proposed approach aligns and discriminates features more precisely, enhancing model robustness against low-quality images.
(2)	The integration of FLCL with Connectionist Temporal Classification (CTC) loss is theoretically sound, leveraging the strengths of both techniques. CTC is widely used in sequence-to-sequence problems like STR, and combining it with frame-level contrastive learning ensures the model learns more robust and discriminative features.

**Limitations:**

(1)	While applying contrastive learning at the frame level is innovative within the context of STR, the concept of contrastive learning itself is not new. To accurately evaluate the novelty of this approach, it is essential to compare it with the current state-of-the-art methods that similarly employ contrastive learning. Such a comparison would provide a clearer perspective on the distinctiveness and potential advantages of the frame-level application within STR.
(2)	The analysis should not be limited to comparisons with CTC-based methods, it should also encompass a broader range of state-of-the-art techniques that have been published in recent years, such as ABINet (2021 ABINet), ConCLR (2022, AAAI), Parseq(2022 ECCV), I2C2W (2023 TPAMI), SSM (2024 IJCAI) et al.

**Suitability:**

2

---

### Official Review · Reviewer_BjAM · 2024-05-28

**Rating:** 4
**Confidence:** 3

**Summary:**

This paper proposes a frame-level contrastive loss for the scene text recognition task. First, the authors calculate contrastive loss using each vector from the feature map. Then, a mamba-based projection module is proposed to replace linear projection in contrastive loss. As a result, the proposed loss outperforms basic contrastive loss on multiple baseline models.

**Strengths:**

1. Employing contrastive learning on a fine-grained level is more suitable for character-level text recognition task.
2. Adequate experiments have been conducted on multiple lightweight STR structures, verifying the superiority of the proposed loss.
3. The paper is well-organized and easy to follow.

**Limitations:**

1. In line 131 and 206, it seems that references [1, 28] in the paper also use frame-level contrastive loss (such as “Frame-to-instance” in SeqCLR). Besides, “Sequence-to-Sequence Contrastive Learning for Text Recognition” (ACM MM2023) also proposes multi-level contrastive learning. Authors should further discuss with them.
2. In line 120, the proposed method cannot reduce the computation or parameters for STR models. I think FLCL has no special advantage for lightweight models. Why do authors only focus on lightweight models and CTC decoders? What is the performance of normal STR models and attention decoders?
3. Transformer layer also can generate bidirectional representation (ABINet, CVPR2021). Can attention-based Text Perceiver get similar results with the mamba layer?
4. Authors should also report their results on Union14M which contains more challenging examples.

**Suitability:**

2

---

### Official Review · Reviewer_1QxF · 2024-05-31

**Rating:** 4
**Confidence:** 3

**Summary:**

The proposed frame-level contrastive learning (FLCL) framework for lightweight Scene Text Recognition (STR) models demonstrates an innovative approach by focusing on frame-level rather than word-level features. The introduction of the Text Perceiver Module (TPM) effectively enhances text-related representations, thereby improving the performance of lightweight models in low-quality scenarios without increasing computational costs. Comprehensive experiments across multiple STR benchmarks and a new low-quality dataset validate the efficacy of the FLCL framework, showing significant performance improvements over existing methods.

**Strengths:**

1. Innovative Approach: The paper introduces a novel frame-level contrastive learning (FLCL) framework for Scene Text Recognition (STR), focusing on frame-level rather than word-level features. This approach is well-written and provides a new direction for improving lightweight STR models.

2. Comprehensive Experiments:
The authors conducted extensive experiments on multiple well-known benchmarks, as well as a new low-quality dataset. The experimental results convincingly demonstrate the effectiveness of the proposed FLCL framework over existing methods. This includes a thorough ablation study and comparisons with other contrastive loss methods.

**Limitations:**

While the FLCL framework presents significant advancements in lightweight STR models, addressing the outlined concerns and inconsistencies will further solidify its contributions to the field.

1. Concerns Regarding Table 6 Experiments and Contribution:
The comparisons with methods that only modify the loss function in Table 6 may be unfair. Integrating the State Space Model (SSM) from Vision Mamba likely contributes significantly to the observed improvements.

2. The "Parameter" Column in Table 5 is not objective:
The current scope of the paper focuses on improving lightweight STR model performance with little or no additional costs. However, using the state space model to replace the linear projection layer should be reflected in the parameter. Including a comparison of computational costs for models with and without the Text Perceiver Module would provide a clearer understanding of the additional overhead associated with the proposed improvements.

3. Some typos and inconsistencies:
* Lines 56-57 contain irrelevant information.
* Lines 326-327 have inconsistent abbreviations.
* Line 408 has a typo in "Mamba."

**Suitability:**

2

---

### Meta-Review · Area_Chair_Wzek · 2024-07-04

**Recommendation:** Accept (Poster)
**Confidence:** 4

**Metareview:**

This paper presents a new approach to enhance lightweight Scene Text Recognition (STR) models by introducing a frame-level contrastive learning (FLCL) framework combined with a Text Perceiver Module (TPM). The proposed method demonstrates improvements in low-quality scenarios, validated through experiments across multiple benchmarks. The reviewers acknowledge the innovative use of frame-level contrastive learning and the effective integration of the TPM, which contributes to the robustness of the STR models. However, some concerns were raised regarding the lack of comparison with closely related works, such as SeqCLR, and the absence of detailed parameter and inference speed analyses.